

# RNGU-NET: a novel efficient approach in Segmenting Tuberculosis using chest X-Ray images

Fuat Turk

Computer Engineering/Faculty of Engineering and Architecture, Kirikkale University, Kirikkale, Turkey

## ABSTRACT

Tuberculosis affects various tissues, including the lungs, kidneys, and brain. According to the medical report published by the World Health Organization (WHO) in 2020, approximately ten million people have been infected with tuberculosis. U-NET, a preferred method for detecting tuberculosis-like cases, is a convolutional neural network developed for segmentation in biomedical image processing. The proposed RNGU-NET architecture is a new segmentation technique combining the ResNet, Non-Local Block, and Gate Attention Block architectures. In the RNGU-NET design, the encoder phase is strengthened with ResNet, and the decoder phase incorporates the Gate Attention Block. The key innovation lies in the proposed Local Non-Local Block architecture, overcoming the bottleneck issue in U-Net models. In this study, the effectiveness of the proposed model in tuberculosis segmentation is compared to the U-NET, U-NET+ResNet, and RNGU-NET algorithms using the Shenzhen dataset. According to the results, the RNGU-NET architecture achieves the highest accuracy rate of 98.56%, Dice coefficient of 97.21%, and Jaccard index of 96.87% in tuberculosis segmentation. Conversely, the U-NET model exhibits the lowest accuracy and Jaccard index scores, while U-NET+ResNet has the poorest Dice coefficient. These findings underscore the success of the proposed RNGU-NET method in tuberculosis segmentation.

# INTRODUCTION

Tuberculosis primarily affects the lungs, and approximately 10 million people have been infected with tuberculosis, according to the World Health Organization (WHO) report published in 2020 (*World Health Organization, 2021*). The simple chest X-ray (CXR) has been widely used for various regions (especially in the low and middle parts of the region). Some of the reasons for the widespread usage of CXR for tuberculosis is its easy accessibility and low price. The early diagnosis of tuberculosis is crucial in reducing the number of tuberculosis patients worldwide (*Lin et al., 2021*). In this sense, medical experts particularly focus on segmenting related regions in CXR. However, the manual segmentation and localization of these cases in CXR are complicated, challenging, and time-consuming. Medical experts may not achieve effective segmentation, and localization

Corresponding author
Fuat Turk, fturk@kku.edu.tr

results in poor resource regions (*Aresta et al., 2019*). To address these shortcomings, some researchers have proposed different automated approaches by employing machine learning and deep learning algorithms (*Kotei & Thirunavukarasu, 2022*; *Kim et al., 2022*). These approaches are proposed for reasons such as helping medical experts diagnose tuberculosis and reducing their workload. The rest of this section explains the proposed segmentation and localization approaches developed using deep learning and machine learning algorithms.

In the literature, many studies have examined the segmentation and localization of tuberculosis using different datasets. A novel cascaded approach was proposed for CXR segmentation by employing a convolutional neural network (CNN) algorithm (*Xue et al., 2020*). Initially, clean samples were selected to train the network, and joint optimization was designed to improve the network performance. Shenzhen Hospital's CXR dataset was used in the study. Although the proposed approach provided 0.9253 accuracy with a noise rate of 25% and 50 epochs, the Dice rate was not presented in the study. In a different study, three datasets were combined for separate feature extraction from ensemble algorithms, including customized VGG16, inception CNN, and residual CNN (*Rajaraman et al., 2018*). The best score was obtained using the Customized VGG16 with a $1 \times 10-4$ learning rate, 0.99 momentum, and $1 \times 10-6$ L2 regularization. A novel extension of the U-Net approach was proposed for a Cross-Manufacturer CXR Segmentation using two different datasets: Montgomery and Shenzhen Hospital CXR datasets (*Zhang et al., 2021*). The proposed approach, DEFU-Net, provided a 0.9154 Dice value for the Shenzhen dataset. COVID-19 segmentation also employed U-Net algorithms using a Kaggle dataset (*Balık & Kaya, 2022*). An accuracy of 0.92 was obtained using the applied U-Net algorithm. A new segmentation-based classification approach was proposed for COVID-19 (*Sharma et al., 2022*). The study had two stages, including segmentation and classification, respectively. U-Net and U-Net+ segmentation models were created using a Kaggle dataset (Chest X-ray Masks and Labels) containing 704 X-ray samples. In another study, a deep convolutional neural network (DCNN) algorithm was created for the Computer-Aided Detection (CAD) of tuberculosis using the Shenzhen Hospital dataset (*Stirenko et al., 2018*). The study achieved 0.78 accuracy for validation with 2,000 epochs. Moreover, accuracy scores for the loss augmentation datasets were calculated using CNN. The CNN algorithm provided the highest accuracy, around 0.74, with 5,000 epochs. On the other hand, the highest score was around 66%, with 8,000 epochs using CNN.

A semantic segmentation model of tuberculous was developed by employing different U-Net model versions (*Rajaraman et al., 2021a*). One version of U-Net (VGG16-CXR-U-Net) had the highest Dice score of 0.5189 and the highest IOU score of 0.3503 in the Shenzhen dataset. Another version of U-Net (VGG16-CXR-U-Net (AT)) had the highest Dice score of 0.7552 and the highest IOU score of 0.6168 in the Shenzhen dataset. *Rajaraman et al. (2022)* evaluated different deep ensemble learning approaches for segmenting tuberculosis, including bitwise AND, bitwise-OR, bitwise-MAX, and stacking. The Shenzhen TB CXR dataset was used in the study, including 2,231 data units for training, 66 for validation, and 33 for testing. The study revealed that the proposed stacked model provided the best statistical scores (IOU: 0.4028, Dice: 0.5743). In the creation of the stacked model, features

were initially extracted from the penultimate layer. Then, a meta-learner model was run in the next layer to combine the extracted features and improve the segmentation process. Five convolutional layers were included in the meta-learner with different numbers of filters in the 1st, 2nd, 3rd, 4th, and 5th convolutional layers (256, 128, 64, 32, and 1, respectively). The lightweight U-Net method was also used for segmentation using the Shenzhen dataset. This approach is the modification of the U-Net model to reduce the latency (*Ngoc et al., 2022*). The employed lightweight U-Net provided a Dice value of 0.7252 for the Shenzhen dataset. A different study tested different U-Net methods using the Shenzhen dataset. The highest Dice (0.5189) and IOU (0.3503) scores were obtained with the VGG16-CXR-U-Net (AT) version (*Rajaraman et al., 2021b*).

Even if previous studies have managed to provide convincing scores based on their objectives, they have some limitations. The main limitation of these studies is that convincing results have not been obtained in terms of performance criteria in segmenting tuberculosis. Another limitation could be the lack of sufficient resource requirements (such as GPU and CPU) to obtain high performance for training and testing processes. The final limitation can be described as difficulties in the implementation of segmentation algorithms due to their complex architectural structures. Considering these limitations, in this study, a new RNGU-NET model is created for the segmentation of tuberculosis. The most substantial innovation that working with the proposed model contributes to science is the updating of the non-local block (NLB) structure. In this context, the following research questions were asked to reveal the effectiveness of the RNGU-NET model in tuberculosis segmentation compared to the U-NET and U-NET+RESNET models.

RQ1: Does RNGU-NET perform better than the alternative algorithms in segmenting tuberculosis?

RQ2: Is RNGU-NET easier to implement compared to the alternative algorithms?

RQ3: Is RNGU-NET an ideal algorithm based on training and resource times which are two of the greatest issues in segmentation problems?

RQ4: Can RNGU-NET show more successful performance metrics with its proposed NLB structure?

The structure of the article is as follows: the proposed novel RNGU-NET model is presented in 'Materials & Methods' with three sub-headings: the dataset, RNGU-NET, and evaluation metrics. 'RNGU-NET' presents the results and discussion on the efficiency of RNGU-NET for the segmentation of tuberculosis. Finally, the conclusion and future directions are outlined.

## MATERIALS & METHODS

### Proposed approach RNGU-net framework

This section provides information about the proposed RNGU-NET architecture. Figure 1 illustrates the general lung segmentation framework. The rest of this section explains each process presented in Fig. 1.

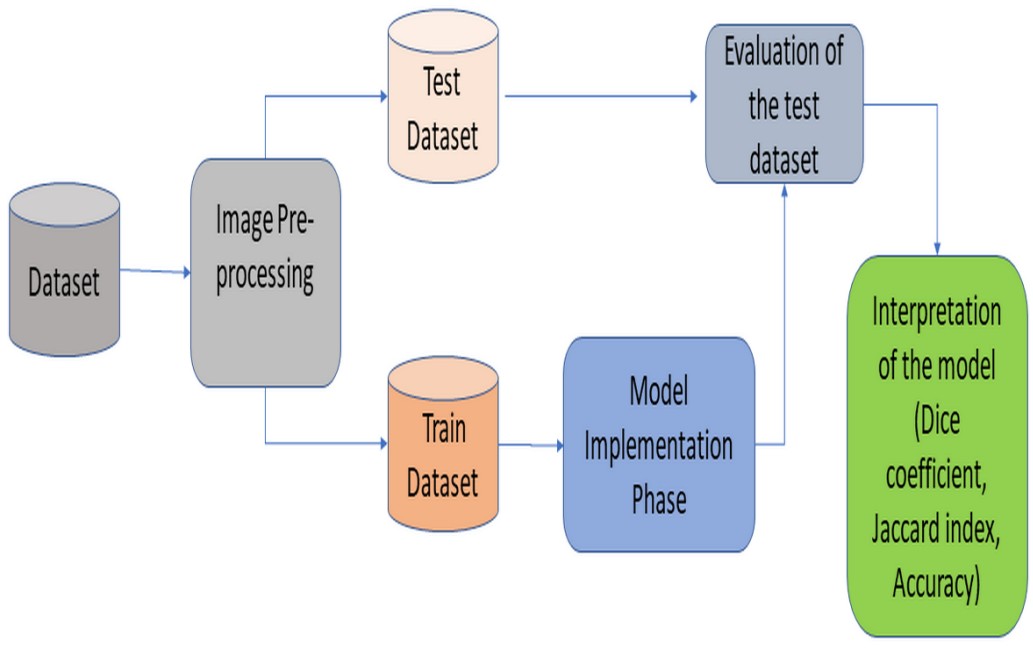

**Figure 1   General lung image segmentation framework.**

## Dataset

The publicly available Shenzhen dataset is used in this study. This dataset consists of CXR images collected in and provided by Shenzhen No. 3 Hospital in Providence, Shenzhen, Guangdong, China (additional descriptions and details about the dataset: available at *Jaeger et al. (2014)*). The dataset has 326 regular and 336 abnormal images (in PNG format) showing symptoms compatible with tuberculosis. The dataset shows a consistent distribution considering the patient classes and normality criteria. Class distribution rates are very close to each other. Moreover, even though the dataset size is relatively small, both qualified studies in the previous literature and the reliability of the institution where the data were collected attracted the attention of the researchers. The Shenzhen dataset includes 1,024 × 1,024 resized images. Additionally, it includes consensus statements for a subset ($N = 68$) from two radiologists for radiology readings (*Jaeger et al., 2014*; *Rajaraman et al., 2021a*). Experienced doctors have proposed segmentation approaches on this dataset, and thus, no additional image pre-processing was applied.

## RNGU-NET

Figure 2 shows the proposed RNGU-NET model for lung segmentation. The architecture of this model is similar to the architecture of the classical U-NET. However, the RNGU-NET model is improved using different architectures in its encoder and decoder phases. ResNet is included in the encoder phase of the model, whereas Gate Attention Block (GAB) is included in the decoder phases of the model (*Zhang et al., 2019a*). The most important difference of the proposed model from classical architectures is its NLB structure, which is

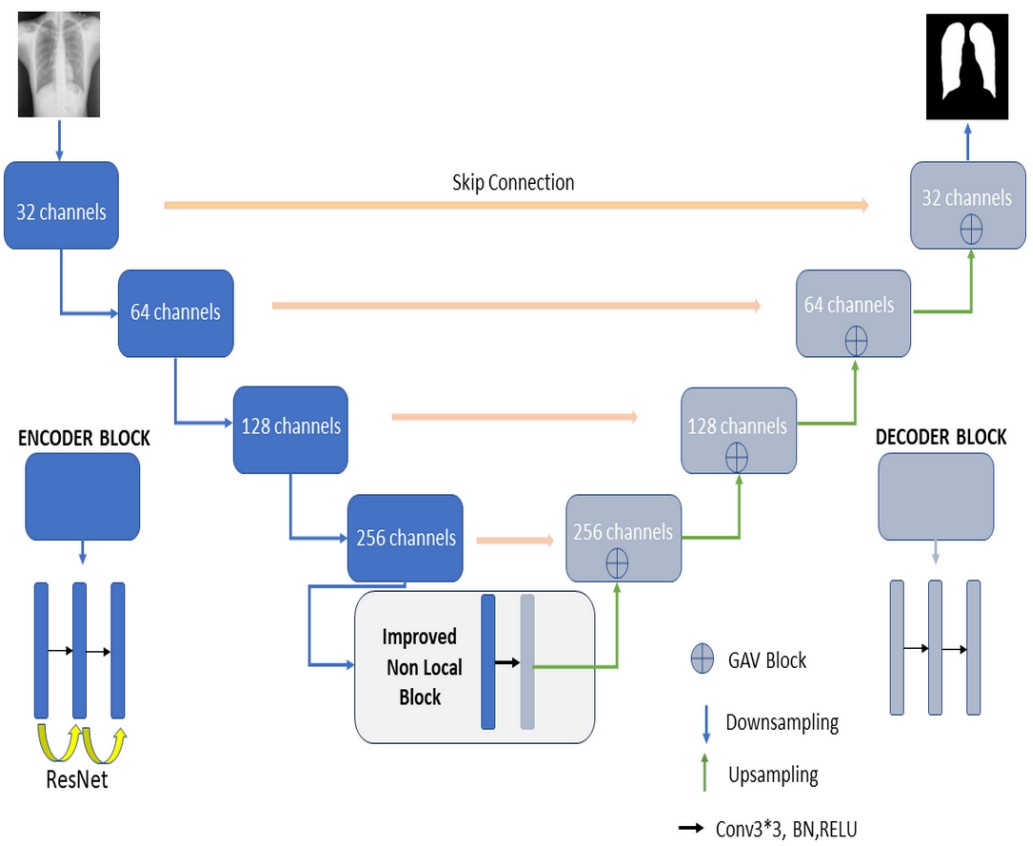

**Figure 2  Proposed RNGU-NET architectural structure.**

improved to solve the bottleneck problem. The architecture of each model block is created using 3 convolutional layers, max pooling, and ReLU activation functions. At the end of the decoder phase, the output is demoted to one dimension with a sigmoid activation function to complete the segmentation process. In Table 1, the layer structures, image sizes, and functions that are used are shown in detail for a better understanding and application of the RNGU-NET model. The number of channel sizes in the 256*256 format is gradually increased from 32 to 256 channels. Convolutional layers are supported in each stage with the ReLU activation function and the ResNet architecture. After the NLB layer reduces the channel size from 256 to 128 in the bottleneck phase, the architecture is gradually reduced again to the output size with the GAB block, and the output is converted to a one-dimensional form with the sigmoid function.

## Encoder phase

In the encoder phase, there are down-sampling layers, including the max pooling layers. There are a total of four block structures, each consisting of three convolutional layers and batch normalization (BN). The ReLU activation function is used in the layers. The Nested ResNet architecture is also used to support the existing block structures. Figure 3 shows the employed ResNet architecture (*Turk, Luy & Barisci, 2021*; *Eckle & Schmidt-Hieber, 2019*).

**Table 1  RNGU-NET model architecture.**

| Layer | Channel size | Operators | Layer | Input size (x) | Operators |
|-------|-------------|-----------|-------|---------------|-----------|
| L-Stage1 | 256*256*(32) | Down conv, ReLU, ResNet | R-Stage1 | 256*128*(128) | Up conv, ReLU,GAV |
| L-Stage2 | 256*256*(64) | Down conv, ReLU,ResNet | R-Stage2 | 128*64*(64) | Up conv, ReLU,GAV |
| L-Stage3 | 256*256*(128) | Down conv, ReLU,ResNet | R-Stage3 | 64*32*(32) | Up conv, ReLU,GAV |
| L-Stage4 | 256*256*(256) | Down conv, ReLU,ResNet | R-Stage4 | 32*32*(32) | Upconv, ReLU,GAV |
| L-Stage5 | 256*256*(256/2) | NLB | R-Stage5 | 1*1*(32) | Conv-sigmoid (output) |

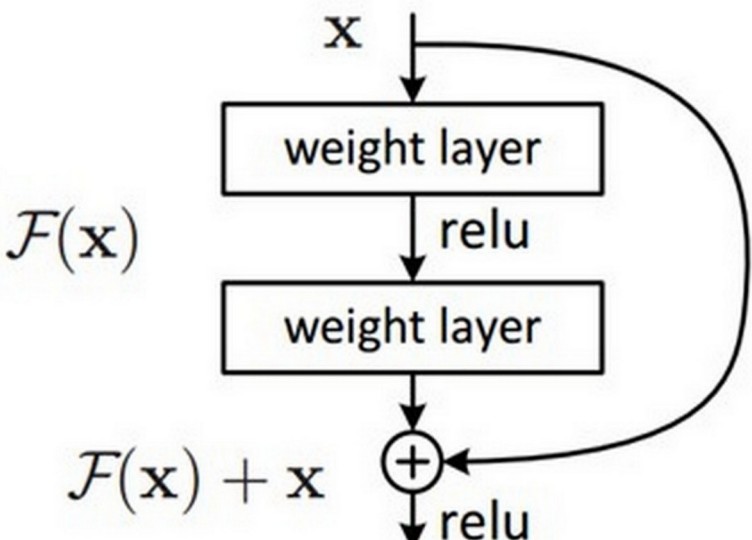

**Figure 3  ResNet architectural structure.**

A nonlinear function could be used to calculate the output. However, arithmetically, the $x$ (input) value is added to the $F(x)$ function by making a shortcut connection from the input to the output. Then, the $F(x)+x$ function is transferred to the ReLU activation function. The values of the previous layers are transmitted to the subsequent layers in a more robust and more stable way by adding the input value at the end of the 2nd layer (*Vununu et al., 2019*).

## Decoder phase

In classical U-Net architectures, channel sizes are reduced in stages in the decoder phase. The model is then executed by creating a relationship between the decoder and the encoder phases, and this process is called skip connection. Extracting the features from the images correctly is essential in this phase. Thus, GAB block structures are integrated into the proposed RNGU-NET model to obtain a more successful decoder phase. It is noted that

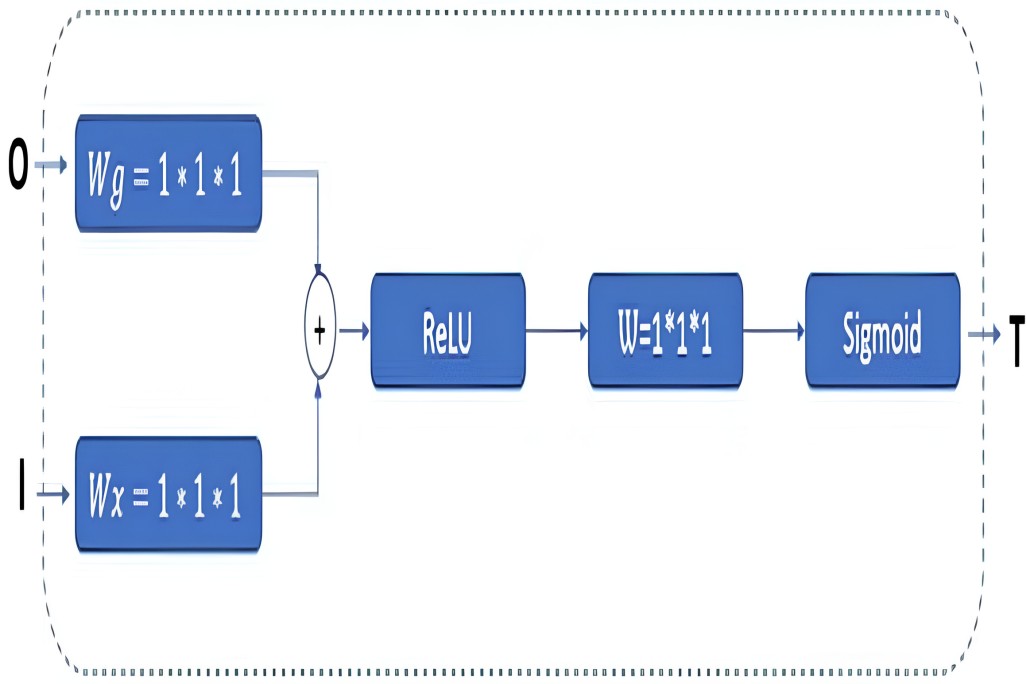

**Figure 4  GAB architectural structure.**

the GAB block structures are generally used with V-Net models. The applied GAB block structure is presented in Fig. 4.

In the GAB block structure, inputs include a guide feature map (I) and a filter feature map (O). Conversely, the output value can be interpreted as a high-resolution feature map called a T shape. In the decoder phase, the attention block is preferred to emphasize the features in the foreground and reduce the effect of the background (*Wang et al., 2018*). The employed GAB block consists of three steps. In the first step, "O" is passed through $1 \times 1 \times 1$ convolution filters based on channels to perform a linear transformation using $C \times H \times W$ feature maps. Then, two transformed feature maps are combined based on element insertion with the ReLU layer. In the final step, the $1 \times 1 \times 1$ convolution block is linearized with a sigmoid activation function to generate the GAB map T (*Zhang et al., 2019b*; *Liskowski & Krawiec, 2016*; *Huang et al., 2020*).

## Proposed non-local block

In the U-NET architecture, the size of the input information is regularly reduced in the encoder phase, starting from the first step. Linear feature representation is learned in the decoder phase, and the size gradually increases. When the decoder phase is finished, the output size must equal the input size. However, since the system's input is compressed linearly, a bottleneck where not all features can be transmitted occurs. Although segmentation architectures are designed to overcome the bottleneck problems in CNNs, the exact transmission of input information is a critical problem. The classical

NLB architecture is used to overcome the bottleneck barrier and minimize data loss (*Wang et al., 2018*). This study redesigns the NLB structure for less feature loss. The Non-Local operation in DNN models is defined as shown in Eq. (1) (*Buades, Coll & Morel, 2005*).

$$y_i = \frac{1}{C(X)} \sum_{\forall J} f(X_i, X_J) g(x_j) \tag{1}$$

Here, $i$ is the index of the output position, and $j$ is the index that enumerates all possible positions. $x$ is the input signal (properties of the inputs such as picture, video, *etc.*), and $y$ is the output signal of the same size as $x$. The binary $f$ function is a scalar calculation function between $i$ and all $j$. The single g function calculates the input signal at position $j$. The response is normalized by factor $C(x)$. In the NLB architecture, feature maps are represented in a tensor format, denoted as T $\times$H $\times$W $\times$1024 for 1,024 channels. "$\otimes$" stands for matrix multiplication, and "$\oplus$" stands for element-wise addition. Red and blue boxes show $1\times 1\times 1$ folds (*Wang et al., 2018*). In Eq. (2), the calculation function for the NLB is shown. $W\_Z$ is the initial value of the weights, $x\_i$ is the connection information, and $y\_i$ shows the exact size information as $x\_i$ and the block value in the $z\_i$ architecture.

$$z_i = W_Z y_i + x_i \tag{2}$$

$W_Z$ is the initial value of weights, $xi\_i$ is the residual connection information, and $xi\_i$ shows the exact size information as $xi\_i$ and the block value in the $z\_i$ architecture.

## Implementation of non-local blocks

In the classical NLB architecture, as in Fig. 5, the number of red boxes (represented channels) is half the number of channels in the input unit, denoted by $X$.

In the proposed architecture, the number of blue boxes is one more than half of the $X$ input (*Kaiming, Zhang & Ren, 2016*). In both architectures, the bottleneck design reduces the computation time of a block by approximately half. The NLB architecture uses $1\times 1\times 1$ convolutions. In general, the softmax function is applied after the first two-convolution matrix multiplication, and the third $1\times 1\times 1$ convolution is again multiplied in matrix size. In the improved NLB architecture, $1\times 1\times 1$ convolutions, one more than half of the input size, are preferred. These convolutions are multiplied as a matrix, and the softmax function is applied. Here, it is possible to extract more features at the same height (H) and same width (W) levels in tensor size (T) with an extra $1\times 1\times 1$ convolution (as seen in Eqs. (1) and (2)), and this process makes the extracted feature map different. After these processes, aggregation is performed based on the elements in both architectures. The classical and improved NLB architectures are shown in Fig. 5.

## Evaluation metrics

Accuracy, recall, precision, F1-score, the Dice coefficient, and the Jaccard Index are used to evaluate the efficiency of the newly proposed RNGU-NET model (*Turk & Kökver, 2022*). The rest of this section presents the formulae and working principles of these calculations. Formulae are presented for accuracy in Eq. (3), recall in Eq. (4), precision in Eq. (5), and

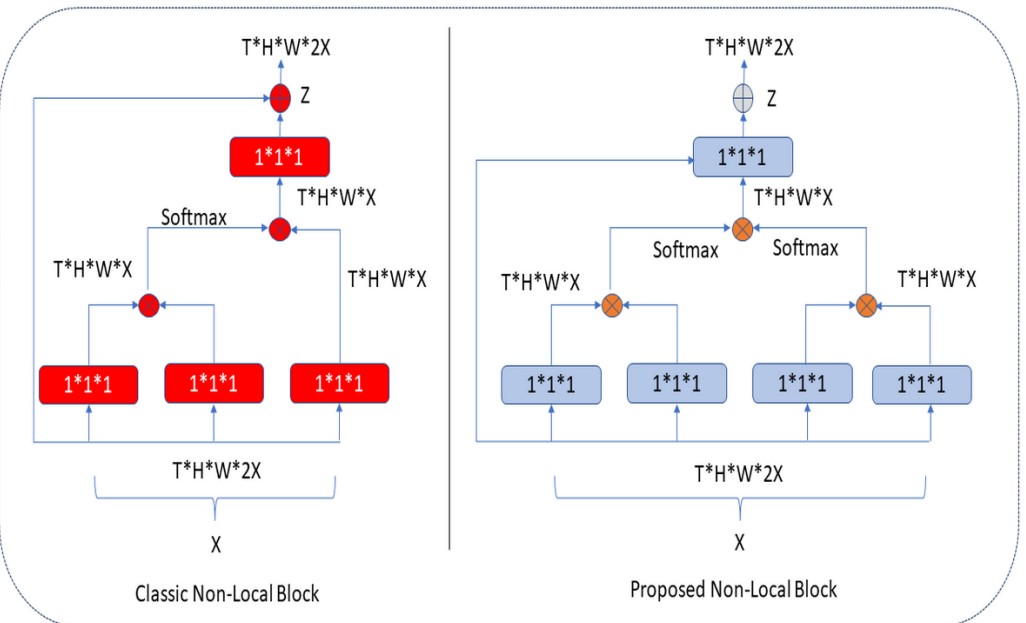

**Figure 5  Classic and proposed NLB architecture.**

F1-score calculations in Eq. (6).

$$Accuracy = (TN + TP)/(TP + FP + TN + FN) \tag{3}$$

$$Recall = TP/(TP + FN) \tag{4}$$

$$Precision = TP/(TP + FP) \tag{5}$$

$$F1.Score = 2 * (Precision * Recall)/(Precision + Recall) \tag{6}$$

The Dice coefficient measures the spatial similarity between two segmentations (*Sudre et al., 2017*). It is widely used to evaluate segmentation performance in the processing of medical images (*Shen et al., 2018*; *Türk, Lüy & Barışçı, 2020*). The Dice coefficient is calculated as shown in Eq. (7). The Jaccard Index, also known as the Jaccard similarity coefficient, is a statistical value used to determine the rate of similarities between sample sets. The measurement highlights the similarity between finite sets of samples and is defined as the intersection size divided by the size of the union of sample sets (*Türk et al., 2022*; *Bouchard, Jousselme & Doré, 2013*). The Jaccard Index is calculated as shown in Eq. (8). The Dice coefficient loss value is accepted as the negative value of the Dice coefficient and is calculated as shown in Eq. (9).

$$Dice\ Coefficient = 2|A \cap B|/(|A| + |B|) \tag{7}$$

$$Jaccard\ Index = |A \cap B|/(|A| + |B| - |A \cap B|) \tag{8}$$

$$Dice\ Loss = -(Dice\ Coefficient) \tag{9}$$

## Training and validation loss/dice coefficient plot

The images in the dataset are first trained and then tested with the U-NET, U-NET+ResNET, and RNGU-NET architectures, respectively. While the classical NLB architecture is used in the U-NET and U-NET+ResNET architectures, the improved NLB architecture is preferred in the RNGU-NET model. The data are randomly distributed into two groups, 80% allocated for training and 20% allocated for testing. The "train_test_split" command in the Python sklearn library was imported for the distribution process. The results in this section are the average of the five-fold cross-validation results obtained from the training dataset. Five sections are run separately, and the average validation result is taken as reference. Thus, a more consistent result is obtained during the training phase. The following parameters are set to the optimum values in the creation of the models. During the training process, the Adam Optimizer is preferred as the optimization algorithm, the filter size is 3*3, the stride size was 2*2, and the padding parameters are chosen as "same". In addition to these parameters, Max pool = 2*2, batch size = 32, input image size = 256*256, and learning rate = 1e−3. The training process takes 50 steps, and early stopping criteria are not applied during this process. However, since it is observed that there is no considerable improvement in the learning process after this step, a longer training duration is not preferred. Finally, the models are run on the Spyder and Jupiter notebook platforms with the TensorFlow-GPU 2.0 library with an Nvidia RTX3060 graphics card. The training processes are completed using the same resources and the same hyperparameters in all three models, and attention is paid to ensure a fairer comparison. In this section, it should be reminded that the U-NET algorithms are an important performance metric for the segmentation task, and the dice similarity coefficient is an important performance metric in the evaluation of these results. The training and validation loss/dice coefficient values obtained as a result of the training process are presented here. The U-NET model is shown in Fig. 6, the U-NET+ResNet model is shown in Fig. 7, and the RNGU-NET model is shown in Fig. 8. In Fig. 6, the training and loss curves show a slight mismatch. In Fig. 7, discrepancies are observed between the curves, especially in the first steps of the training process. The training and validation curves are in convincing harmony in Fig. 8 based on the training and validation results with RNGU-NET. It is seen that this compatibility is directly proportional to the improvements in the encoder and decoder stages of our (RNGU-NET) model.

## Test scores of the employed models

Table 2 presents the test scores of the employed models when the epoch value is set as 50. As seen in Table 2, the best performance is obtained with the new RNGU-NET with 98.56% accuracy, 97.21% Dice coefficient, and 96.87% Jaccard Index values. Additionally, other performance metrics (recall: 98.75%-precision: 97.64%) also support the success of the model. Therefore, the proposed RNGU-NET model is a segmentation model with a positive response to research question 1 (RQ1). Even though the contribution of the RNGU-Net model seems small when looking at the performance metrics, small positive increases are valuable since the performance results of all models can be considered successful. Furthermore, it should not be forgotten that small folds and image differences will be

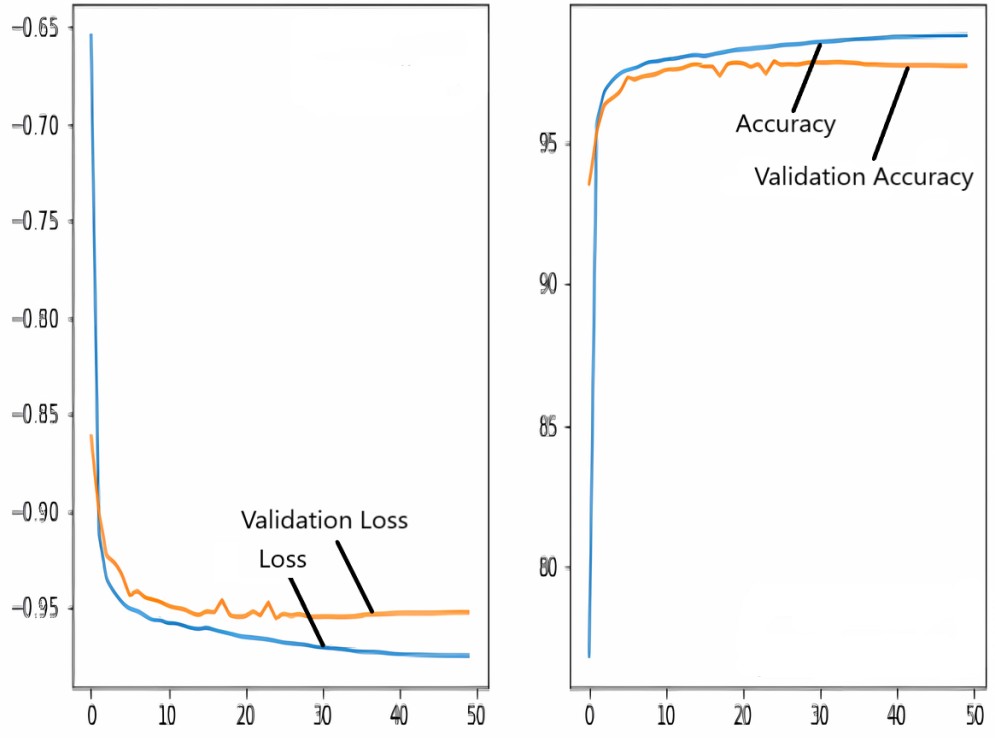

**Figure 6  Loss/dice coefficient graph of U-NET for training and validation.**

detected at earlier stages during the segmentation process. As seen in the application stages, another positive aspect of the model is that it can be easily implemented like the classical U-NET model. Therefore, the RNGU-NET model, which provides a positive solution to the 2nd research question (RQ2) of this study, can be implemented in a shorter time than its alternatives (other complex models such as V-Net architectures).

## Evaluation scores of RNGU-NET

Figure 9 shows the model evaluation results of U-NET, while Fig. 10 shows the model evaluation results of U-NET+ResNet. Based on the information in both figures, the results can be considered satisfactory when the original images are compared to the mask images obtained from the model. Figure 11 shows the evaluation results of the RNGU-NET model. In the comparison of the original images and mask images obtained using the proposed model, the results seem highly successful and convincing. It can be observed that the proposed RNGU-NET model succeeds in segmenting almost the whole image area accurately, except for minor edge differences. In Fig. 12, the random results obtained from the operation of all three models are presented comparatively. Although the RNGU-NET model appears to be more successful in segmenting the original image, the success of the U-NET model is also quite good. The U-NET+ResNet model segmented only a certain region unsuccessfully (the region selected with the red ring). Finally, in Fig. 13, we show the segmentation results, especially those of U-NET and RNGU-Net. Results on images

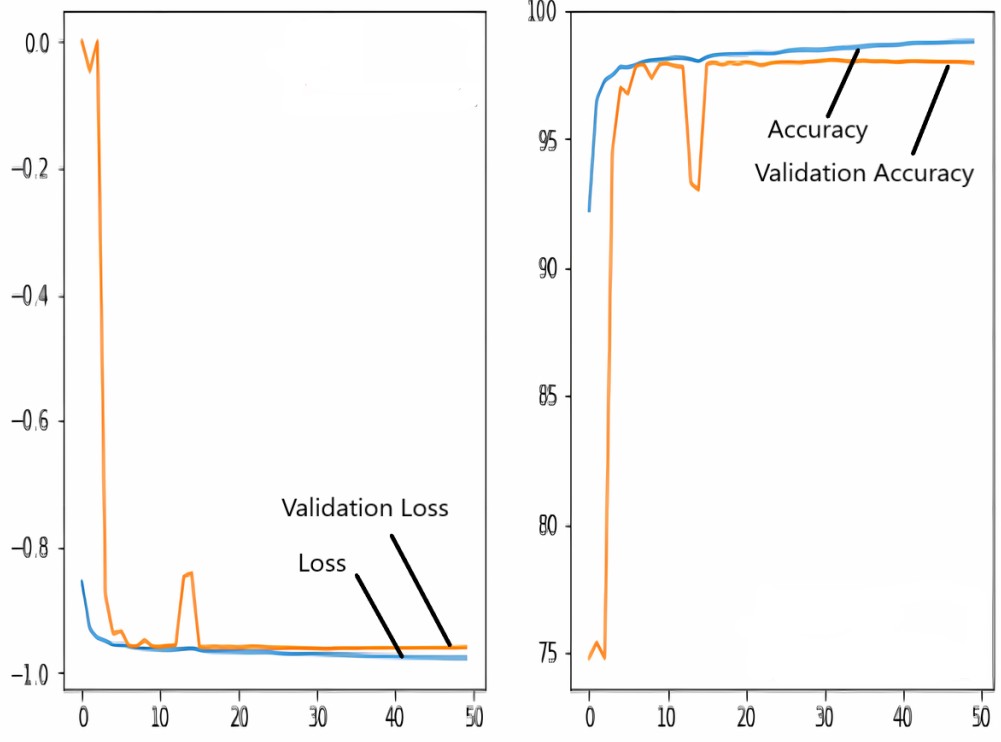

**Figure 7** **Loss/dice coefficient graph of U-NET+ResNet for training and validation.**

that are more difficult for the model are shown. Here, while the U-NET and RNGU-NET models fail at small curves and narrow edges, the U-NET+ResNet model shows a good performance for easier segmentation. These results prove once again how difficult task segmentation is.

## General comparison of studies on lung segmentation and localization proposed by different researchers

In the DEFU-NET architecture, fusion and densely connected recurrent convolution block structures are used in the feature encoder and feature decoder stages (*Zhang et al., 2021*). However, this architecture requires additional processing and resources. Therefore, the training process for DEFU-NET is getting longer. On the other hand, the proposed RNGU-NET architecture is designed with a more straightforward structure, as presented in Fig. 2. Nevertheless, using the Shenzhen dataset, the RNGU-NET model provides a better statistical score (0.9708) than DEFU-NET (0.9154). In a different study (*Sharma et al., 2022*), the classical U-NET architecture for segmentation was used with the basic CNN model for classification in COVID-19 diagnosis using CXR scans. Considering the Dice coefficient and accuracy results based on the proposed RNGU-NET, the positive modification of the U-NET model is seen in Table 3. Although some other studies also achieved high levels of success, it should be emphasized once again that a positive contribution of 1–2% to the results is valuable for noticing small details in the segmentation
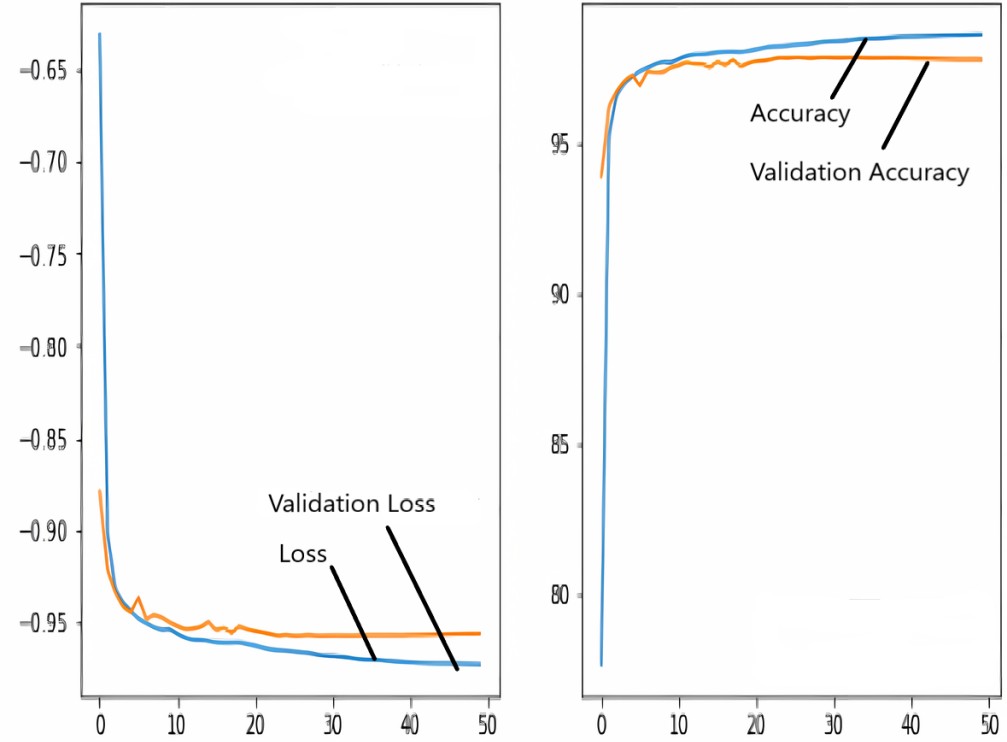

**Figure 8** Loss/dice coefficient graph of RNGU-NET for training and validation.

**Table 2  Test evaluation results of U-NET models.**

| Model | Acc. | Recall | Precision | F1 Score | Dice coef. | J. Index |
|---|---|---|---|---|---|---|
| U-NET | 97.76 | 98.09 | 97.51 | 97.80 | 96.51 | 95.71 |
| U-NET+RESNET | 98.27 | 98.49 | 97.46 | 97.97 | 96.33 | 96.03 |
| RNGU-NET | 98.56 | 98.75 | 97.64 | 98.03 | 97.21 | 96.87 |

process. *Rajaraman et al. (2021b)* conducted a study for the segmentation of tuberculosis by employing the VGG16 and VGG19 models. Although these models successfully solved classification problems, the statistical results obtained for segmentation problems were not at the desired level. While the model was tested on the Shenzhen and Tuberculosis X-ray (TBX11K) datasets, it can be argued that the results need to be improved to obtain convincing scores. A new segmentation model was proposed for the segmentation of tuberculosis by using the Lightweight U-Net model (*Ngoc et al., 2022*). The Lightweight U-Net model focused on standard up-convolution and skip connections in the study. Although this model had a simple structure, the obtained accuracy result was 72.52%. For this reason, it can be thought that improvements (such as the ResNet block) should be made in the model.

In general, the Shenzhen dataset has been used in the training and testing processes of segmentation models presented in the literature. Thus, the new RNGU-NET model is also

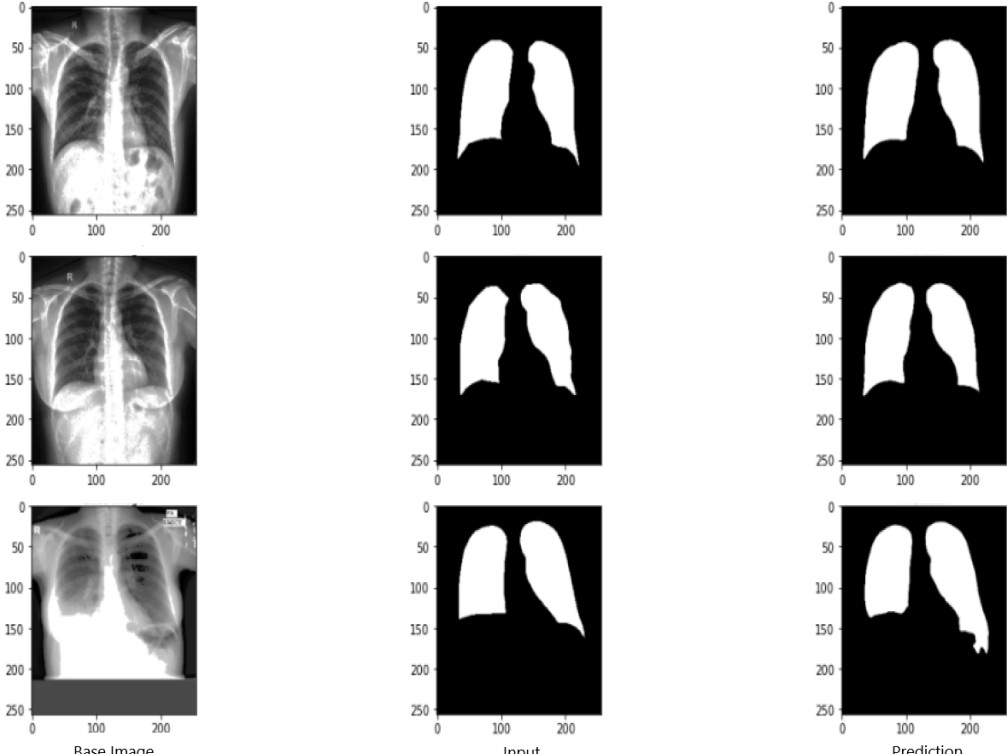

| Base Image | Input | Prediction |

**Figure 9  Evaluation images for U-NET model.** X-ray image source credit: *Jaeger et al., 2014*, Shenzhen Hospital CXR Set.

trained and tested with the Shenzhen dataset to reveal its efficiency in the segmentation of tuberculosis compared to alternatives. At this point, the main differences between the RNGU-NET model and currently available models are the proposed NLB architecture, the application of the solution for bottleneck bypass, and the modifications made in the encoder and decoder phases of the U-NET model. As seen in Table 3, the proposed RNGU-NET provides a Dice coefficient of 97.21% and 98.56% accuracy through the modifications. The obtained scores confirm the efficiency of the RNGU-NET model in the segmentation of tuberculosis, which shows a positive response to research question 3 (RQ3) of this study.

Table 4 presents statistical information about the employed U-NET, U-NET+ResNet, and RNGU-NET models regarding the training time, resource usage, and epoch/batch size parameters. In many segmentation problems, the number of layers increases, and nested and complex block structures are used for performance improvement (*Xia et al., 2021*). In this case, the rate of resource usage increases (*e.g.*, more CUDA requests or memory requirements), and the training time is longer. It is not a preferred situation due to costs and time loss. Additional block structures added to the layers during the improvement process of segmentation actually place a substantial burden on system resources. However, according to the results in Table 4, this load is very low with the proposed model, and this affects the performance results positively. These results prove the effectiveness of the model regarding the issue raise in research question 4 (RQ4).

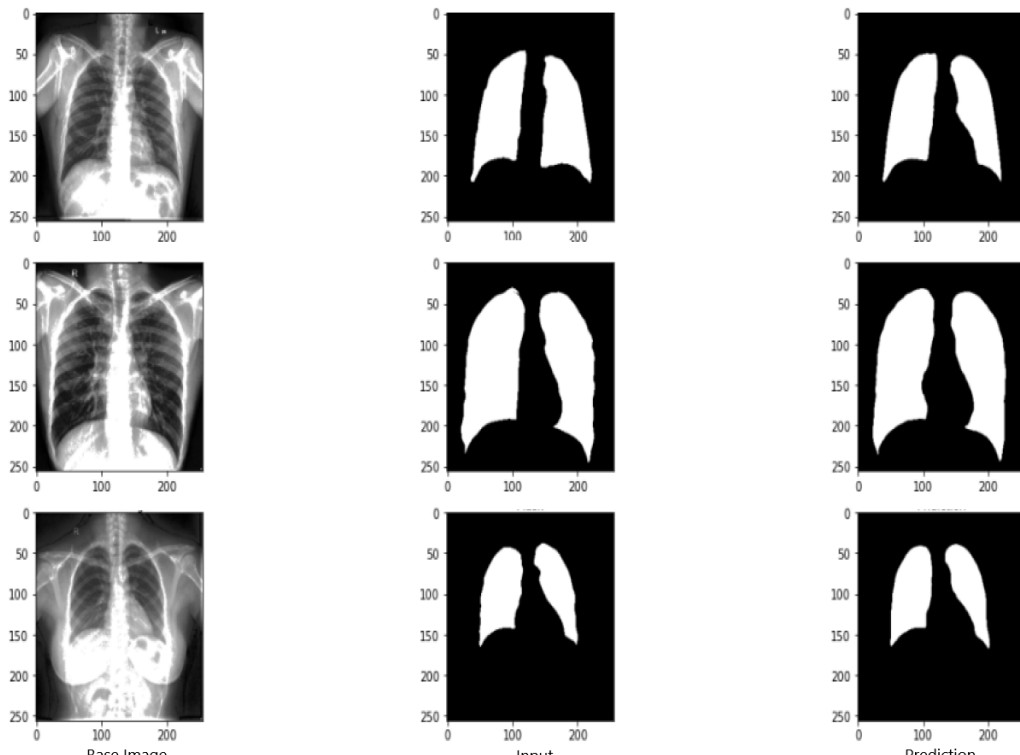

**Figure 10** **Evaluation images for U-NET+ResNet model.** X-ray image source credit: *Jaeger et al., 2014*, Shenzhen Hospital CXR Set.

## CONCLUSION

In this study, a new segmentation model, called RNGU-NET, is proposed for tuberculosis segmentation. The efficiency of the proposed model is measured using the prestigious Shenzhen dataset. The model manages to transfer the ResNet block and convolutional layers to each other only in the encoder phase. Then, GAB blocks, primarily used in V-NET models, are successfully integrated into the decoder phase. With the improved NLB architecture, the bottleneck problem in the segmentation process is seen to improve significantly. This study reveals that the RNGU-NET model provides promising results in the segmentation of tuberculosis. As a limitation, it should be noted that despite its successful results, the RNGU-NET model should be tested on other datasets. In future studies, an evaluation of the proposed model is planned in datasets or patient images collected from different regions. This is very important for testing the effectiveness and reliability of the model. Moreover, it would be more beneficial to improve the models proposed for segmentation with incremental modifications, by avoiding complex structures as much as possible. It is deemed appropriate to evaluate the model with real data by creating manual masks over images (with support from expert radiologists) in all areas, especially in medical image processing studies where segmentation is difficult. Thus, I am optimistic

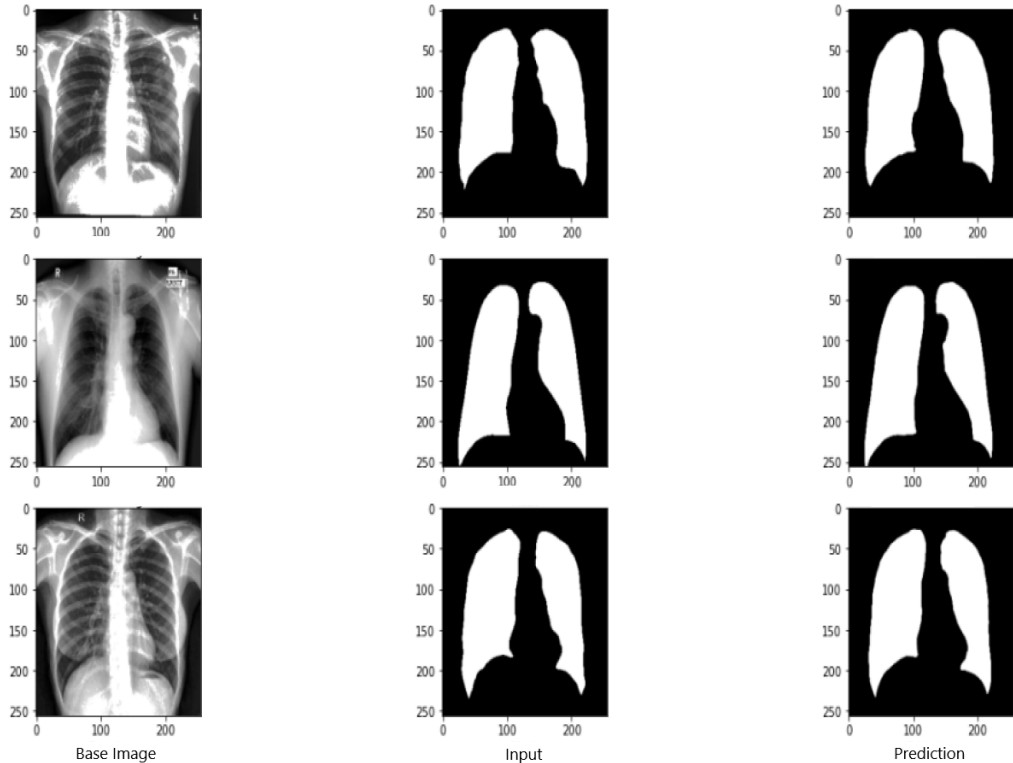

**Figure 11  Evaluation images for RNGU-NET model.** X-ray image source credit: *Jaeger et al., 2014*, Shenzhen Hospital CXR Set.

that effective work can be done at lower cost and using a lower rate of system resources.

### Funding
The authors received no funding for this work.

### Competing Interests
The authors declare there are no competing interests.

### Author Contributions
- Fuat Turk conceived and designed the experiments, performed the experiments, analyzed the data, performed the computation work, prepared figures and/or tables, authored or reviewed drafts of the article, and approved the final draft.

### Data Availability
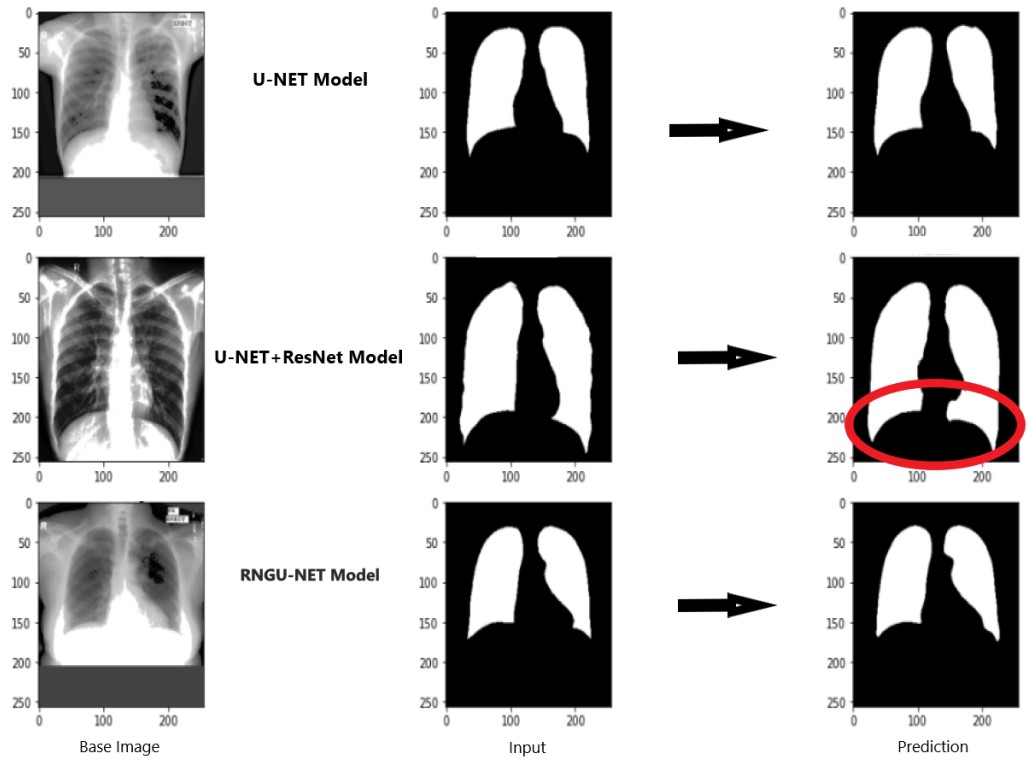

**Figure 12  Evaluation images for three model (easy segmentation).** X-ray image source credit: *Jaeger et al., 2014*, Shenzhen Hospital CXR Set.

**Table 3  Studies on lung segmentation and localization proposed by different researchers.**

| Study | Dataset | Model | Dice Coef. | Accuracy |
|---|---|---|---|---|
| *Xue et al. (2020)* | Shenzhen | Sample Selection + Joint Optimisation | – | 0.9253 |
| *Rajaraman et al. (2018)* | -Shenzhen -Montgomery -Private | Customized VGG16 | – | 0.917 |
| *Zhang et al. (2021)* | -Shenzhen (S) -Montgomery (M) -Combination S + M | DEFU-Net | 0.9154 0.9227 0.9667 | – – 0.9804 |
| *Balık & Kaya (2022)* | -Kaggle Dataset | U-NET | – | 0.92 |
| *Sharma et al. (2022)* | Kaggle Dataset (Chest Xray Masks and Labels) | U-NET U-NET+ | 0.9488 0.9235 | 0.9635 0.9610 |
| *Stirenko et al. (2018)* | Shenzhen | DCNN | | 0.74 |
| *Rajaraman et al. (2021a)* | Shenzhen Tuberculosis X-ray (TBX11K) | VGG16-CXR-U-Net VGG16-CXR-U-Net (AT) | 0.5189 0.7552 | - - |
| *Rajaraman et al. (2022)* | Shenzhen | Stacked Ensemble | 0.5743 | – |
| *Ngoc et al. (2022)* | Shenzhen | Light-weight U-Net | 0.7252 | – |
| **Proposed model** | **Shenzhen** | **RNGU-NET** | **0.9721** | **0.9856** |

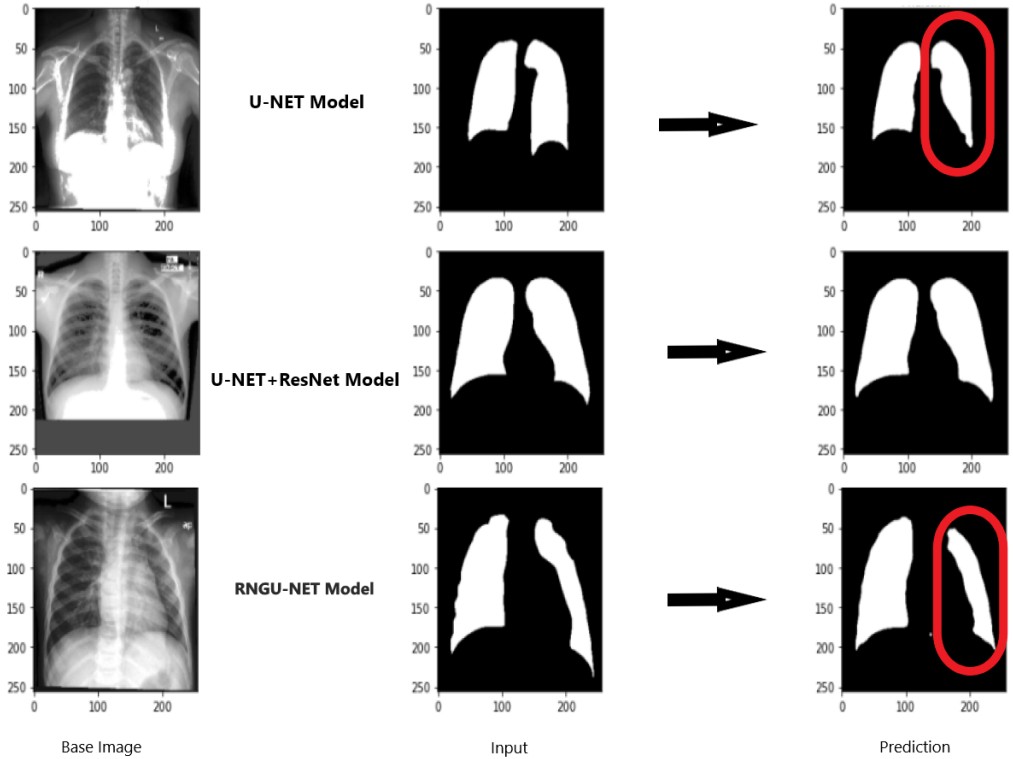

**Figure 13** **Evaluation images for three model (difficult segmentation).** X-ray image source credit: *Jaeger et al., 2014*, Shenzhen Hospital CXR Set.

**Table 4** **Statistical information about the employed segmentation models.**

| Model | Training time (min) | Resource usage | Epoch/batch size |
|---|---|---|---|
| U-NET | 48 | RTX3060 | 100/32 |
| U-NET+ResNet | 51 | RTX3060 | 100/32 |
| RNGU-NET | 56 | RTX3060 | 100/32 |

The codes are available in the Supplemental File. The dataset used is the Shenzen dataset from the National Library of Medicine. Access to that dataset can be requested here: Available at https://openi.nlm.nih.gov/faq#faq-tb-coll.

## Supplemental Information
Supplemental information for this article can be found online at http://dx.doi.org/10.7717/peerj-cs.1780#supplemental-information.

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
