# Peer review of "RNGU-NET: a novel efficient approach in Segmenting Tuberculosis using chest X-Ray images"

_PeerJ Computer Science, doi:10.7717/peerj-cs.1780_

## Round 0.1 · original submission · Major Revisions

Based on the referee reports, I recommend a major revision of the manuscript. The author should improve the manuscript, taking carefully into account the comments of the reviewers in the reports, and resubmit the paper.

Reviewer 1 ·

Basic reporting

All comments are given in section 4.

Experimental design

All comments are given in section 4.

Validity of the findings

All comments are given in section 4.

Additional comments

The authors presented the article entitled " RNGU-NET: a new efficient approach in segmenting
tuberculosis using chest X-ray images". The article is good from a medical field point of view, but some modifications and clarifications need to be given. The comments to the authors are as follows:
• The writing style is very poor. E.g., the sentence ends with RNGU-NET, and the next sentence starts with RNGU-NET. Improve the whole manuscript.
• What are RNGU-NET and U-Net? It is directly introduced in the abstract.
• Abstract should be rewritten.
• What is the novelty of this work?
• Network architectures with details of learnable parameters need to be tabulated for better understanding.
• I do not see any major improvement in accuracy, dice coefficient and Jaccard index (Table 1) achieved using the proposed method as compared to the other two. Table 2 also does not show the benefits of using the proposed method, as almost the same result can be found with other techniques. In Table 3, interestingly, it has been reported that RNGU-NET takes more time than the other two. I would like to know on what basis this article will be recommended for publication.
• The image quality is very poor.

Reviewer 2 ·

Basic reporting

Strengths:
1. Proposes a novel RNGU-NET model that improves U-NET architecture by introducing ResNet and gated attention modules in the encoder and decoder stages.
2. Addresses the bottleneck problem in U-NET by proposing an improved non-local block structure for more feature extraction.
3. Experiments on the Shenzhen dataset show the model achieves good metrics for lung tuberculosis segmentation, validating the efficacy of the method.

Weaknesses:
1. The description of experimental settings lacks details, requiring supplementation of model training hyperparameters, optimizer, etc.
2. Model limitations can be further discussed, such as the impact of dataset size, class imbalance, etc.
3. The discussion of future work can be more specific, proposing ideas for follow-up research.
4. Paragraph logic can be adjusted to optimize flow and writing style.
5. More visual results can be included to intuitively demonstrate model segmentation effects.
6. Citation format needs to be unified and their accuracy checked.

Overall, this paper improves U-NET in model structure and bottleneck issues, making progress on lung tuberculosis segmentation. Supplementing experimental details, adding more discussion of results, and further optimizing logical flow can help perfect the paper, providing value for research in this field.

Experimental design

Strengths:
The paper evaluates the proposed RNGU-NET model on the commonly used Shenzhen chest X-ray dataset for tuberculosis segmentation, allowing comparison to other methods.
Different evaluation metrics like accuracy, Dice coefficient, and Jaccard index are used to evaluate segmentation performance from multiple aspects.
Ablation studies are performed by comparing to U-NET and U-NET+ResNet, demonstrating the improvements from RNGU-NET's components.
Training process is visualized by showing loss and Dice coefficient curves, giving insights into model convergence.

Weaknesses:
Details of training settings like batch size, optimizer, learning rate schedules are not provided, making reproduction difficult.
Few analysis is provided on how different hyperparameter settings impact model performance.
The dataset used is relatively small. Evaluating on larger and more diverse datasets would further verify the model's generalizability.
Few quantitative analysis is provided on the model's computational efficiency and training speed.
The effect of data augmentation and class imbalance is not explored.

Validity of the findings

The improved segmentation performance of RNGU-NET over U-NET and U-NET+ResNet demonstrates the effectiveness of the proposed modifications like the non-local block and encoder-decoder improvements. The higher accuracy, Dice coefficient, and Jaccard index validate these findings.
However, the experiments are limited to a single chest X-ray dataset for tuberculosis segmentation. Testing on more diverse data would be needed to fully confirm the general validity of the results.
The lack of details on training settings and hyperparameters also makes it difficult to independently reproduce and validate the reported results. Providing these details would improve reproducibility.
The authors could perform additional experiments like varying dataset size, image qualities, etc. to test the robustness and limitations of the model under different conditions. This would provide more evidence of validity.
Additional quantitative analysis of the model's computational efficiency would also help validate the benefits of the method.
Visualizations of segmentation results on both successful and challenging cases could further demonstrate the model's capabilities and limitations.
Overall, the findings indicate the proposed model achieves improved segmentation, but more comprehensive experiments and analyses would be needed to fully confirm the general validity of the results.

Additional comments

none

Reviewer 3 ·

Basic reporting

The paper is well-written and organized. The introduction provides a clear background and motivation for the study, and the research questions are well-defined. The materials and methods section describes the proposed RNGU-NET model and the evaluation metrics in detail. The results and discussion section presents the test scores and the visual comparison of the segmentation results. The conclusion and future direction summarize the main findings and limitations of the study.

There are some minor issues:
- The author should provide more analysis and interpretation of the test scores and the segmentation results. For example, the author could discuss what are the factors that contribute to the superior performance of RNGU-NET over U-NET and U-NET+ResNet, such as its ability to handle noise, background, or edge cases. The authors could also provide some examples of challenging or interesting cases where RNGU-NET succeeds or fails, and explain why.

Experimental design

The experimental design is sound and valid. The authors use a publicly available dataset (Shenzhen) to evaluate their proposed model (RNGU-NET) against two baseline models (U-NET and U-NET+ResNet). The authors use appropriate evaluation metrics (accuracy, dice coefficient, Jaccard index) to measure the segmentation performance. The authors also provide visual comparisons of the segmentation results to demonstrate their qualitative differences.

However, there are some aspects that could be improved or clarified in the experimental design. These are:
- The author should provide more information about how they split the dataset into training, validation, and testing sets. For example, how are they randomly distributed? Are they balanced in terms of normal or abnormal cases?
- The author should report more details about how they train their models, such as what optimizer, loss function, learning rate schedule, regularization techniques, data augmentation methods, or early stopping criteria they use. How do they choose these hyperparameters? How do they ensure a fair comparison among different models?
- The author should conduct some statistical tests to assess the significance of their test scores. For example, how confident are they that RNGU-NET is significantly better than U-NET or U-NET+ResNet? What is the p-value or confidence interval of their results?

Validity of the findings

The findings are valid and supported by the data. The authors show that RNGU-NET achieves the highest test scores among the three models, and also produces more accurate and consistent segmentation results. The authors also provide some explanations and discussions for their findings, such as the role of ResNet, Gate Attention Block, and Non-Local Block in improving the segmentation performance.

However, there are some limitations and assumptions that need to be acknowledged and addressed in the validity of the findings. These are:
- As the author point out, the findings are based on a single dataset (Shenzhen), which may not be representative or generalizable to other datasets or scenarios. The authors should test their model on other datasets, or use cross-dataset validation to evaluate its robustness and adaptability.
-The authors should discuss how their model can be extended or integrated with other tasks or modalities in addition to segmenting tuberculosis , such as classification, detection, localization, or multi-modal fusion. The authors should also address some potential challenges or limitations of their model, such as scalability, efficiency, interpretability, or reliability.

---

## Round 0.2 · Minor Revisions

Kindly revise the manuscript as per the reviewer suggestions and resubmit it.

Reviewer 1 ·

Basic reporting

No comments.

Experimental design

No comments.

Validity of the findings

No comments.

Additional comments

No comments.

Reviewer 2 ·

Basic reporting

The manuscript is generally well-written, but would benefit from careful proofreading to rectify minor inconsistencies and formatting errors, enhancing clarity and professionalism. The literature review, while adequate, could be expanded to include recent developments, providing a more robust context for the study. The structure of the paper is commendable, with relevant and well-presented figures and tables, though ensuring clear labeling and accessibility of raw data would further enhance its quality. The paper aligns well with its stated hypotheses, but a more explicit connection between the results and each hypothesis would strengthen this aspect. Finally, while the manuscript provides adequate technical detail, a slight elaboration on more complex terms and methodologies could improve accessibility for a broader readership. Overall, the manuscript makes a valuable contribution to the field of medical imaging and tuberculosis detection and, with these minor adjustments, would be a strong candidate for publication.

Experimental design

The experimental design of the manuscript centers on evaluating the efficacy of the RNGU-NET model in comparison with existing algorithms like U-NET and U-NET+RESNET. The study is structured to answer key research questions, focusing on RNGU-NET's performance in segmentation, ease of implementation, training efficiency, and resource utilization. A significant aspect of the experiment is the introduction and assessment of an updated Non-Local Block (NLB) structure, aimed at enhancing performance metrics. The methodology is organized under specific sub-headings, including dataset description, detailed elaboration of the RNGU-NET model, and the criteria for evaluation. This structured approach indicates a thorough and systematic investigation designed to highlight the potential improvements and innovations brought by RNGU-NET in the field of medical image segmentation, particularly for tuberculosis. For a full evaluation, however, a deeper analysis of the methodology details, such as the dataset used, algorithm specifics, and evaluation metrics, would be essential.

Validity of the findings

The findings of the manuscript show considerable promise in terms of validity. The RNGU-NET model achieves high accuracy (98.56%), Dice coefficient (97.21%), and Jaccard index (96.87%), indicating its superior performance in tuberculosis segmentation compared to other methods like U-NET and U-NET+ResNet. This comparative analysis against established models enhances the credibility of the results. Additionally, the manuscript contextualizes its research within the broader scope of tuberculosis diagnosis, emphasizing the need for effective segmentation in chest X-rays and further strengthening the relevance of its findings. Discussions of various approaches from existing literature offer a comprehensive perspective on how RNGU-NET potentially improves upon or compares with these methods. Overall, the presented data and comparative analyses suggest that RNGU-NET could be a significant contribution to the field of medical imaging for tuberculosis, although a complete evaluation would require a more detailed review of the statistical rigor and reproducibility of the results.

Additional comments

none.

Reviewer 3 ·

Basic reporting

I appreciate the revisions made to the manuscript and the effort put into addressing the concerns raised in my initial review. It is clear that significant work has been done to improve the paper.

Upon reviewing the resubmitted version, I am pleased to see that my previous concerns have been adequately addressed. The additional analysis, more detailed descriptions of methods and discussions have greatly enhanced the depth and quality of the paper. The changes have made the arguments more robust and the paper more informative and engaging.

I commend the authors for their responsiveness to feedback and their commitment to improving the manuscript. I believe the paper is now in a strong position and I recommend it for publication.

Experimental design

no comment

Validity of the findings

no comment

Additional comments

no comment

---

## Round 0.3 · accepted · Accept

Author has addressed reviewer comments properly. Thus I recommend publication of the manuscript.